# Trampolining Accidents in an Adult Emergency Department: Analysis of Trampolining Evolution Regarding Severity and Occurrence of Injuries

**DOI:** 10.3390/ijerph19031212

**Published:** 2022-01-22

**Authors:** Nora Sasse, Mairi Ziaka, Lara Brockhus, Martin Müller, Artistomenis K. Exadaktylos, Jolanta Klukowska-Rötzler

**Affiliations:** 1Department of Emergency Medicine, Inselspital, Bern University Hospital, Bern University, 3010 Berne, Switzerland; norasasse@gmail.com (N.S.); lara.brockhus@outlook.com (L.B.); martin.mueller2@insel.ch (M.M.); aristomenis.exadaktylos@insel.ch (A.K.E.); 2Department of Internal Medicine, Thun General Hospital, 3600 Thun, Switzerland; mairi.ziaka@gmail.com

**Keywords:** trampoline, trampolining accidents, fractures, adults, retrospective study

## Abstract

Purpose: Only a few studies have been conducted on trampoline-associated injuries in adults, especially in Switzerland. The aim of the present study was to describe the characteristics of trampoline-related injuries in patients older than 16 years of age and track their development over time by comparing two different time periods. Methods and Materials: Data were gathered from the emergency department (ED) of Bern University Hospital. A retrospective design was chosen to allow analysis of changes in trampolining accidents between 2003–2020. Results: A total of 144 patients were enrolled. The number of ED presentations due to trampoline-related injury rose significantly over time. The most common injuries were sprains to the extremities (age group 31–62: 58.4% and age group 16–30: 50.8%), followed by fractures (age group 31–62: 33.3% and age group 16–30: 32.5%). Lower extremities appeared to be the most frequently injured body region (age group 31–62: 20.8% and age group 16–30: 10.0%), although the differences were not statistically significant, *p* = 0.363. BMI was significantly higher for older than for younger patients (*p* = 0.004). Conclusion: Over the last two decades, trampoline-related injuries have become more common in patients older than 16 years of age. These are most common in the lower extremities. While most of the patients in the present study only suffered minor injuries, the occasional severe injury might result in long-term disability. As trampoline-related injuries in adults are becoming more common, prevention strategies in public education and safety instructions must be optimised.

## 1. Introduction

Trampolining has gained great popularity [1] since its invention by George Nissen in 1936. It was used for pilot training in World War II, while in 1948, the first national championships were held and in the year 2000, the first Olympic competition started [2]. Therefore, trampolining as a competition and school sport has developed into a fashionable hobby and the number of commercial trampoline parks has grown worldwide. For example, the number of trampoline parks in Switzerland has risen from three in 2009 to over 1000 in 2017 [3]. Moreover, trampolining has a variety of health benefits. In children, trampolining improves mental development and functional performance and increases bone strength [4,5]. As regards adults, a recent study showed that patients with type 2 diabetes benefit from trampoline activity by improving insulin resistance, lipid profile, and waist circumference, and thus can serve as a useful non-pharmacological intervention to decrease cardiovascular risk factors and boost performance in patients with type 2 diabetes [5]. Moreover, it has been convincingly demonstrated that mini-trampoline exercise may improve dynamic stability and balance in older persons by increasing plantar flexor muscle strength and the ability to regain balance during forward falls [6]. 

Nevertheless, trampolining also has downsides, as shown by studies on trampolining accidents in children [3,4,7,8,9,10,11,12,13]. Indeed, several studies demonstrate that there has been an increase in the incidence of trampoline-related injuries in children in recent decades [14]. Although the majority of injuries are minor, approximately 10% of all paediatric trampoline-associated injuries are severe [8], which can ultimately lead to major functional disability such as paraplegia and quadriplegia or even death [1,7,15,16]. Trampolining injuries in adults have not been adequately investigated [17,18,19,20,21], although severe injuries such as head and neck trauma may lead to death or long-term disability [8,17,22]. In addition, a recent study examined trampoline-related injuries in adults and found a three-fold increase in surgical interventions compared to children [23]. As demonstrated by previous studies, the most common mechanisms of trampoline-associated injuries are a fall on or out of the trampoline and collision with other persons [1,3,4,18]. In view of the increasing number of accidents caused by trampolining, many studies have aimed to address safety measures which should be implemented to reduce the possibility of injuries [4,17,24,25,26].

In light of the above, in the present study, we aimed to investigate the epidemiology of trampoline-related injuries and identify their underlying mechanisms, their causes, and trauma patterns in order to facilitate the development of simple but efficient preventive measurements on the basis of the results. 

## 2. Materials and Methods

### 2.1. Study Design

We conducted a retrospective data analysis which comprised patients older than 16 years old who had presented to the Inselhospital adult emergency department (ED) between 2003 and 2020 with a trampoline-associated injury. No individual informed consent was obtained. The procedure for the descriptive case series was approved by the cantonal (i.e., district) ethics committee in Bern (No.: 2018-01958). 

### 2.2. Exclusion Criteria

Patients without a general consent (GC) were excluded from the study. In addition, patients who actively refused a GC were excluded. Moreover, children under 16 years of age were also excluded, as they were admitted to a separate emergency department. Cases of patients with diffuse pain symptoms and no direct relation to trampolining were not included in the analysis. The case of a patient who was presented to the ED with severe polytrauma after being hit by a trampoline flying through the air due to stormy weather was additionally ruled out. 

### 2.3. Data Collection and Retrospective Data Analysis 

The present retrospective study encompassed data from patients older than 16 years of age who were admitted to the adult emergency department due to traumatic trampolining injuries between 2003 and 2020. The collected data were evaluated by gender, age, year, season, weekday, body mass index (BMI), treatment area, triage, route of admission, route of discharge, site of accident, mechanism of injury, type of injury, anatomical location of injury, type of trauma, and costs. The population was divided into two age groups: young adults aged 16 to 30 and older adults aged 31 to 62. We used two age groups in order to identify the trends in those specific age groups and to better distinguish accidents related to school/university facilities and injuries associated with indoor recreation halls. The costs were categorised into 3 groups: CHF 1 to 1000, 1001 to 10,000 and >10,000 (Swiss francs). For BMI, patients were divided into 4 groups: no information, <18.5, 18.5 to 25 and >25 kg/m^2^. The treatment area included the different medical departments, that is, surgery, walk-in clinic, neurology, orthopaedics, ophthalmology, craniomaxillofacial surgery, ENT (i.e., ear, nose, throat), and internal medicine. Depending on the severity of the injuries, the patients were triaged from 1 to 4, with 1 being the most urgent. Because of lack of information, we had to add a 5th category for “no information”. The mode of arrival was categorised as no information, self-admission, other hospital, family doctor, ambulance or Swiss air-rescue, and discharge modes were categorised as no information, discharged, hospitalised or transferred to another hospital. Additionally, the location of the accident was investigated, including school sports, trampoline centre, and private property; however, the location of the accident was often missing, due to underreporting in the original medical files. 

For mechanisms of injury, we included nine subgroups, that is, no information, misstep on/beside the trampoline, impact on/beside the trampoline, distortion, contusion with self or second involved person, collision with additional object on the trampoline or no active trauma. Moreover, types of injury included 7 subcategories: no information, contusion, sprain, fracture, soft tissue lesion, neurological deficits, and concussion. The fractures were again categorised into the different body parts: upper/lower extremities, cervical/thoracic and lumbar spine, face/skull, and clavicle. The aforementioned subcategories were also applied for the category “location of injury”. Finally, the type of trauma was distinguished between mono-trauma, combined trauma, and polytrauma.

### 2.4. Statistical Analysis

The data were summarised using descriptive statistics (mean values, percentages). The statistical analysis was performed using Stata® 13.1 (StataCorp, The College Station, TX, USA). The distribution of categorical variables is given, with the absolute number and the relative number as a percentage. The clinical parameters were compared between the two age groups (16–30 and 31–62), the sex (male/female), and the two groups for year of consultation (2003–2011 and 2012–2020). Categorical variables were analysed using the chi square (χ^2^) test and the Fisher exact test. The threshold of significance was set at *p* < 0.05 (two tailed).

## 3. Results

### 3.1. Demographic Characteristics

Between 2000 and 2020, a total of 144 patients with trampoline-related injuries were identified in our database. Of all patients presenting with trampoline-associated injuries, 61.1% were male and 83.3% were between 16 to 30 years of age. Appendix A illustrates the comparison between the two age categories, namely 16 to 30 and 31 to 62 years of age. Older patients (31 to 62 years old) had a significantly higher BMI (*p* = 0.004), with a BMI value > 25 kg/m^2^ in 25% of the cases. Furthermore, younger patients were more often discharged home (*p* = 0.006). The location of the accident in the older group was more often on private property than during school sports (29.2%; *p* = 0.005). Finally, we recorded higher costs for older patients (31 to 62 years old), that is, 41.7% between CHF 1001–10,000 (in comparison with 26.7% for younger patients) and 12.5% for more than CHF 10,000 (in comparison with 5.8% for younger patients). The lowest cost category—of up to CHF 1000—was dominated by the younger patients (16 to 30 years old), with 40.8% (in comparison with 37.5% in the older age category).

The comparison between genders is presented in Appendix A. The mode of admission differed significantly between male and female patients. Relatively more females were admitted through ambulance transfer (14.3%) or air ambulance (8.9%) (*p* = 0.011), whereas male patients were more frequently identified as walk-in patients (57.1%). Fractures were more often observed in females than males, corresponding to 39.3% and 28.4%, respectively (Figure 1), and therefore were triaged as more urgent.

### 3.2. Mechanisms and Pattern of Injury

With a roughly 48% incidence, lower extremities were the most commonly injured anatomical location, although the differences were not statistically significant. For both men and women, the second most frequently injured anatomical region was the head and the cervical spine (Figure 1).

Figure 2 highlights the distribution of different types of injuries. The most common category of injury was sprain to the extremities (43.2% for males and 37.5% for females), followed by fractures (28.4% for males and 39.3% for females). For both genders, fractures were most common in the lower extremities. As regards the mechanisms of injury, 27.1% involved a misstep on or next to the trampoline, followed by collision next to the trampoline in 24.3% of the patients (Figure 2). No statistically significant difference was observed between males and females.

### 3.3. Trends

An increase in the total number of trampoline-related injuries was seen in the second study period (between 2012 and 2020; Appendix A). In the time period from 2003 to 2011, 33 patients were admitted to the emergency department due to a trampolining accident, whereas in the period from 2012 to 2020, 111 patients were admitted (i.e., the gradual increase is shown in Figure 3). The incidence of trampoline-related injuries in patients older than 36 years of age tends to increase from 3% for the period 2003–2011 to 13.2% for the period 2012–2020. A decrease in injuries related to school facilities was observed for the second period of the study (24.2% vs. 14.4%) and was accompanied by an additional increase in indoor recreation halls (i.e., 3% to 12.6%; Appendix A). Additionally, our data indicated decreasing incidence of fractures and traumatic brain injuries over time. In the two time periods, that is 2003 to 2011 and 2012 to 2020, the percentage decreased from 40% to 29.8% for fractures and from 5% to 2.9% for traumatic brain injuries (one case of incomplete traumatic quadriplegia). 

As mentioned above, older patients (31 to 62 years old) showed significant higher BMI when compared to the younger age group (i.e., 16–30 years old). A statistically significant increase in BMI was observed between the two study periods. Thus, BMI was greater than 25 kg/m^2^ in 12.1% of the cases for the period 2003–2011 and in 16.2% of the cases for the period 2012–2020 (*p* < 0.001). With respect to the Swiss Triage Scale, we found a significant association between the two study periods. The group of patients with urgent triages was greater between 2012 and 2020—64% of the patients—when compared to the corresponding percentage for the study period 2003–2011 (15.2%; *p* < 0.001). In reference to discharges, our data showed that, when a new treatment area called “FastTrack” was introduced to ED, more patients with less severe injuries were discharged home, whereas in the earlier time period, patients with the same injuries were treated through the surgery department, resulting ultimately in more hospitalizations than in the second time period of the study. More specifically, for the study period 2003–2011, 30.3% of the patients were hospitalised, while for the study period 2012–2020 this parameter was significantly reduced (i.e., 12.6%, *p* = 0.046). Moreover, for the two different study periods, the incidence of fractures and traumatic brain injuries tended to decrease (i.e., from 40% to 29.8% and 5% to 2.9%, respectively).

## 4. Discussion

Since its invention in the 1930s, trampolining has become a popular sport—not only among children and adolescents, but also among adults. As a consequence, trampolining is associated with a significant increase in injuries. Although the majority of trampoline-related injuries are minor, approximately 10% of all paediatric trampoline-associated injuries are severe [8], which can ultimately lead to major functional disability or even death [1,7,16]. In contrast to the paediatric population in which characteristics of trampoline-related injuries are well documented [3,4,7,8,9,10,11,12,13], little is known about the data for adults [1,17,18,19,20,21].

The present retrospective study is an analysis of the progression of trampolining accidents in adolescents and adults, taking into consideration different and various aspects such as gender, age, BMI, route of admission, triage urgency, mechanism of injury, type of injury, anatomical location, and identification of potential risk factors. In contrast to pre-existing studies, in which both adult and paediatric patients were included, the advantage of the present study is that it focuses only on patients older than 16 years of age by quantitatively summarising a large number of data related to trampoline-related injuries [1,18,19].

One of the main findings of our study is a twelve-fold increase in trampolining accidents over the time period of 2012–2020 in comparison to the time period 2003–2011 (i.e., 11 patients versus 133 patients). It is striking that the incidence of trampoline-related injuries in patients older than 36 years of age increased from 3% for the period 2003–2011 to 13.2% for the period 2012–2020. This finding is in accordance with previous observations, which demonstrate increased trampoline popularity since 2012 in adults [26,27], which is further accompanied by the rise of trampoline parks [27]. Our findings additionally demonstrate a decrease in injuries related to school facilities in the second study period (24.2% vs. 14.4%), while the trampoline injuries associated with indoor recreation halls seemed to rise from 3% to 12.6% for the same study period. Furthermore, our data indicated that the number of patients with urgent triages significantly increased between 2012–2020 when compared to 2003–2011 (64% vs. 15.2%, respectively). This last finding could be attributed to the development of a new treatment area for patients with less-severe injuries called “FastTrack”, especially when the percentage of patients discharged home for the same time period is additionally taken into account. Finally, it was shown that the younger group of patients is more often discharged home whereas the older group more frequently requires hospitalisation. In order to understand and interpret this finding, it is important to consider that trampolining requires high demands on the muscles of the lower extremities [28,29] and that the most common pathogenetic mechanism of falls, especially in the elderly, is loss of stability in the forward direction [30]. Indeed, trampolining requires a combination of several components, such as strength, body stability, muscle coordination, and spatial integration [6], parameters which are negatively affected in older individuals [31,32,33]. Moreover, even though trampoline use has standard safety measures such as padded frames [14], inexperienced adult users may choose not to follow safety instructions—potentially leading to serious injury [14,17]. On the other hand, it has been clearly demonstrated that mini-trampoline exercise may improve dynamic stability and balance ability in older persons by increasing the strength of plantar flexor muscles and the ability to regain balance during forward falls [6]. Thus, our findings highlight the need for safety and education measures—not only for children, but also for adults.

Our study also found differences between the type of injuries and their anatomical location [1,18,34]. Although the differences were not statistically significant, injuries to the lower extremities were observed in approximately 48% of the cases and were the most common site of injury in both genders, followed by injuries to the head and cervical spine. Despite the fact that the majority of the patients of the present study sustained minor head and spinal cord injuries (e.g., overstretching of soft tissues), spine fractures were also observed in fourteen patients and cerebral haemorrhage and right vertebral artery dissection were documented in two additional patients (data not shown). This observation is in accordance with previous findings showing an association between trampolining and spine and/or head injuries [35]. Hyperflexion and hyperextension have been considered as plausible underlying causes for cervical spinal cord injuries, resulting from violent landing on the trampoline after undertaking a somersault, and irrespective of its direction (i.e., backward or forward) [15,36,37]. In addition, findings show that a person engaged in trampolining can achieve heights of up to 9 m and has an increased risk of trauma if landing on the head or neck [7], leading potentially to severe and permanent neurological disabilities. Although many studies discuss the unpredictable nature of trampoline-related spine injuries and the inadequacy of even improved safety measures to fully prevent them [22], examination of trampoline-associated accidents and their possible long-term effects on quality of life should encourage potential users to actively consider their participation and to take the risks into account. Moreover, ongoing research in this area contributes to the refinement of the currently existing safety measures. 

Although previous studies reported that fractures were the most common type of injury [1,18,34], we found that sprains to the lower extremities were the most common injury, followed by fractures (i.e., 41% and 32.6% of the cases, respectively). An additional notable finding that emerged in the present study concerns the anatomical location of the fractures. Specifically, and contrarily to most paediatric studies in which fractures to the upper extremity were identified as the most common injury related to trampoline activity [1,3,4,7,24], the present study in adults showed a higher incidence of fractures in the lower extremities, namely in 11.8% of the cases, in comparison to fractures of the upper extremities (4.9%). Nonetheless, in accordance with our findings, Doty et al. (2019) reported a higher incidence of fractures to the lower extremities in both paediatric and adult populations for accidents occurring in jump parks [23]. 

Our most striking finding may be the decrease in the incidence of fractures and traumatic brain injuries over time. For the two time periods, that is, 2003–2011 and 2012–2020, the percentage proportion for fractures decreased from 40% to 29.8% of the cases, and for traumatic brain injuries from 5% to 2.9% (one case of incomplete traumatic quadriplegia). This decrease could be attributed to the increased security/safety measures adopted for trampolining during the latter time period after studies highlighted and identified its potential risks [4,17,24,25,26].

The present study aimed to improve the analysis of the mechanism of injuries resulting from falls while trampolining. Until now, a number of studies have demonstrated that the most common causes of trampoline-related injuries are falls on or out of the trampoline and collisions with other people [1,3,4,18]. However, in our study restricted to adults, 27.1% of the injuries were caused by a misstep on or next to the trampoline followed by an impact next to the trampoline in 24.3% of the cases. This may indicate that the injury pattern may change with age and that additional safety measures may be needed in adults.

One of the most interesting findings of our study is that older patients (31 to 62 years old) had a significantly higher BMI. Particularly, for older patients, BMI was greater than 25 kg/m^2^ in 25% of the patients when compared to the younger age group (i.e., 16 to 30 years old); to the best of our knowledge, this is the first study identifying obesity as a significant risk factor for trampoline-associated injuries in patients between 31 and 62 years old. Although the above finding is not surprising, if we take into account studies showing that obesity may be associated with an increased risk of falls-specially in people older than 60 years of age [38]-a data-based relationship between trampolines, obesity, and injuries has been missing. The underlying causes for the observed finding seem to vary. While it is proposed that obesity is related to sedentary behaviour, medication use, and chronic morbidities, there are several mechanistic factors which predispose obese patients to falls. These factors include postural disability and impaired postural control, poor lower-limb muscle quality, and increased foot loads [38], and these should be taken into account when trampolining is chosen as a preferred sports activity. Moreover, as stated above, obesity is associated with chronic comorbidities, such as obstructive sleep apnoea, diabetes mellitus, hypertension, heart disease, stroke, and others [39], which can increase the risk of instability and result in falls. Indeed, complications related to the above-mentioned conditions-such as daytime hypercapnia and sleep disordered breathing-might result in excessive fatigue due to obstructive sleep apnoea [40] and side effects of medication that enhance the risk of trampoline-related injuries. On the other hand, obesity may have a potential protective role in trampoline-related injuries. For example, three studies in patients with motor vehicle accidents have demonstrated that patients with obesity showed a reduced incidence of head injuries [41,42,43]. However, the data of our study cannot provide a firm conclusion regarding safety measures for trampolining. Further studies are needed to evaluate obesity as a risk factor related to trampolining-associated injuries in order to improve prevention strategies and safety instructions. 

Despite the strengths of the present study, some limitations should be taken into consideration. Specifically, the present study was a single-centre retrospective study and relied on medical records including costs, BMI, mechanism, and location of accident that were not always complete. Additionally, our study did not include long-term follow up information, which could be of interest regarding the health costs and persistence of the disability identified when presented at the ED (i.e., short-term vs. long-term). Moreover, for some patients, data were incomplete or missing. 

## 5. Conclusions

Although the relationship between trampolining and injuries has been well studied in children, there is less relevant information in adolescents and adults, even though this might depict different patterns and highlight additional safety needs. The present study tried to fill this gap by showing that the increased popularity of trampolining in adults in recent years resulted in a significant increase in trampoline-related injuries for adolescents and adult patients. It was additionally demonstrated that, although the majority of injuries in patients older than 16 years of age are minor, severe injuries may still result in long-term disabilities. Therefore, intensive prevention strategies in public education and safety instructions for professional use must be optimised, especially in the context of the continuously growing trampoline parks, in which people without proper training, experience, and, more importantly, of different ages, may be engaged in trampolining as a fashionable hobby or, worse, as a one-time fun activity.

## Figures and Tables

**Figure 1 ijerph-19-01212-f001:**
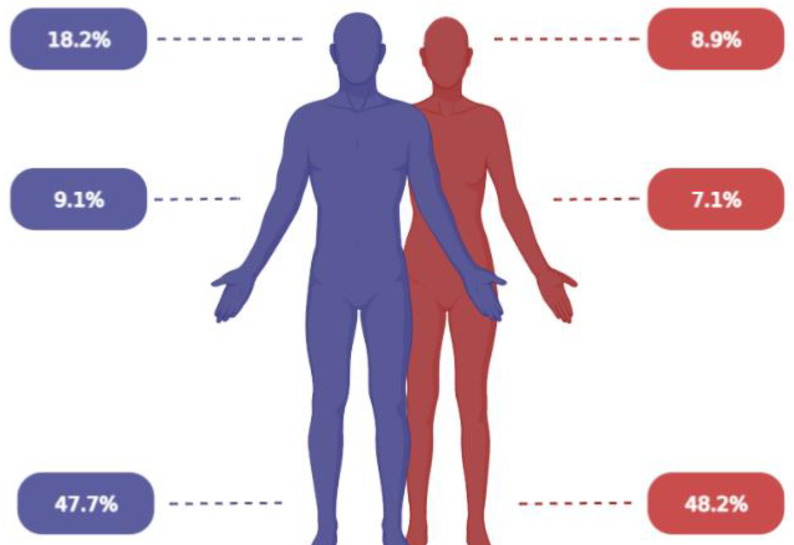
Illustration of injury region between genders. Figures created with BioRender.com.

**Figure 2 ijerph-19-01212-f002:**
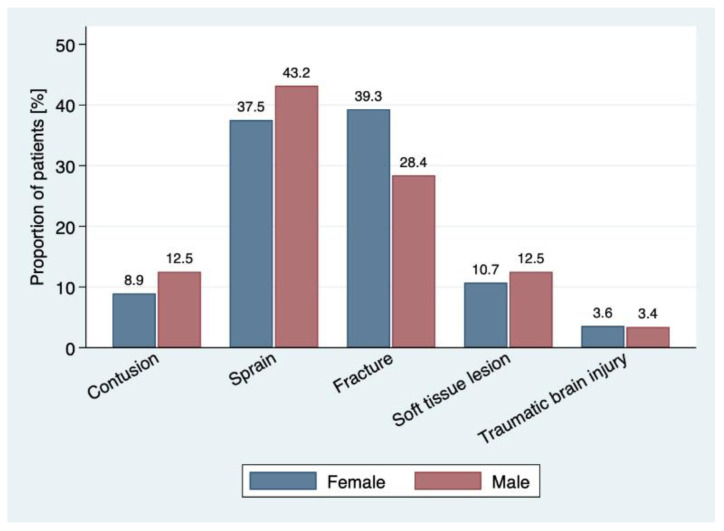
Distribution of different types of injuries.

**Figure 3 ijerph-19-01212-f003:**
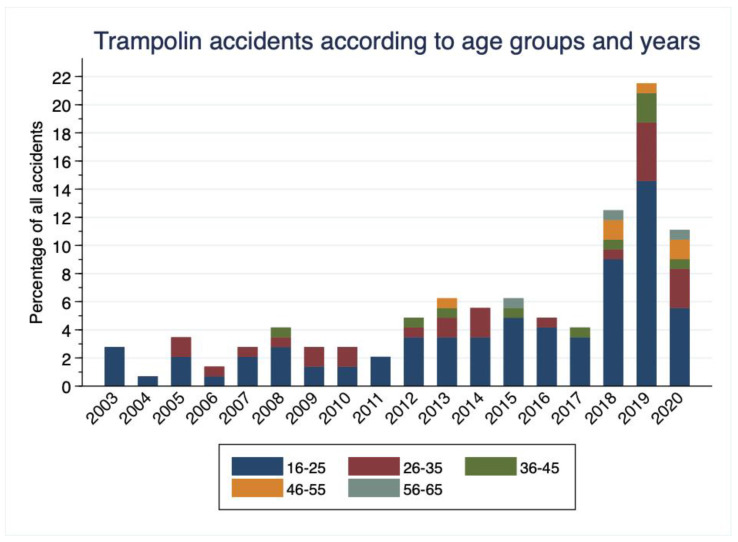
Trampoline accidents according to age groups and years.

## Data Availability

Not applicable.

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
