# Peer review of "Trampolining Accidents in an Adult Emergency Department: Analysis of Trampolining Evolution Regarding Severity and Occurrence of Injuries"

_ijerph, 2022, doi:10.3390/ijerph19031212_

Round 1
Reviewer 1 Report
Dear Authors,
This is a very interesting and well written study about a context world spread and emergente that need to be addressed and prevent. This paper raises awareness for the increase in trampoline injuries rising over time. Given the rise in the use of trampoline social activities this is a very timely paper in term of relevance.
I only have a few comments to help improve your manuscript:
Line 38: in my opinion the term "non-pharmacological intervention" is more consensual in scientific context than "non-medication therapy".
lines 110 and 126: in the sentence and in the legend referred to table 1 “…namely 16 to 30 and 31 to 63 years of age…”. In the table you had wrote 62, and not 63 years. Please rectify.
line 268: in the sentence “These factors include reduced postural disability” you probably meant “reduced postural stability” or "postural disability” … please rectify.
line 269: in the sentence “…poor lower-limb muscle quality…” you probably meant …” poor lower-limb muscle function”.
Additionally I suggest the careful review of all typing errors, the inclusion of numbering in the list of references.
Author Response
Statement Review 1
We would like to sincerely thank you for your extensive comments, inputs and questions. The review helped a lot to further improve our paper. We hope that the changes meet your requirements.
Since many comments were related to the comparison of our data with international studies, we have written a general statement below:
Line 38: in my opinion the term "non-pharmacological intervention" is more consensual in scientific context than "non-medication therapy".
Reply: Done as recommended.
lines 110 and 126: in the sentence and in the legend referred to table 1 “…namely 16 to
30 and 31 to 63 years of age…”. In the table you had wrote 62, and not 63 years. Please rectify.
Reply: We thank the reviewer for this recommendation. In light of the useful recommendation the right age of the population is now stated in the manuscript.
line 268: in the sentence “These factors include reduced postural disability” you probably meant “reduced postural stability” or "postural disability” … please rectify.
Reply: Done as recommended.
line 269: in the sentence “…poor lower-limb muscle quality…” you probably meant …” poor lower-limb muscle function”.
Reply: Done as recommended.
Additionally I suggest the careful review of all typing errors, the inclusion of numbering in the list of references.
Reply: We have reviewed our manuscript carefully and corrected typos.
Reviewer 2 Report
The manuscript is overall sound but I have several comments:
- English needs minor adjustments throughout the text
- 'extremities distortion' has to be rephrased to something from the general orthopedics terminology; I assume the authors mean 'sprain'
- the inclusion and exclusion criteria can be better structured
- the pictured talus dislocation is not relevant to the manuscript and should be removed.
Thank you
Author Response
Statement Review 2
We would like to sincerely thank you for your extensive comments, inputs and questions. The review helped a lot to further improve our paper. We hope that the changes meet your requirements.
Since many comments were related to the comparison of our data with international studies, we have written a general statement below:
- English needs minor adjustments throughout the text
Reply: We have reviewed our manuscript carefully and corrected typos.
- 'extremities distortion' has to be rephrased to something from the general orthopedics terminology; I assume the authors mean 'sprain'
Reply: We thank the reviewer for the very important comment. In light of the edifying recommendation we have now rephrase the terminology.
- the inclusion and exclusion criteria can be better structured
Reply: We thank the reviewer for bringing the above issue to our attention. We have revised the criteria to address the issues raised.
- the pictured talus dislocation is not relevant to the manuscript and should be removed.
Reply: Done as recommended.
Reviewer 3 Report
Review of manuscript
Trampolining accidents in adults: analysis of trampolining evolution in adults regarding severity and occurrence of injuries
It is interesting article concerning important topic, but need some corrections.
Title suggest that adults population will be analyzed, but in the study inclusion criteria were above 16- years of age. It should be change to over 18, and additional statistical analysis should be conducted.
In text citation style should be uninfected. Both are used normal and upper numbers.
Introduction is very short and does not included review of literature: what we already know form previous studies.
Why the border of group selection was age 30? How authors justify it?
104 line in mu opinion p<0.05 instead of p=0.05.
Results. I would recommend instead of 3 similar huge tables presentation only of statistically significant results.
In discussion results are repeated but not discussion this other authors was conducted.
Author Response
Statement Review 3
We would like to sincerely thank you for your extensive comments, inputs and questions. The review helped a lot to further improve our paper. We hope that the changes meet your requirements.
Since many comments were related to the comparison of our data with international studies, we have written a general statement below:
Review 3
Title suggest that adults population will be analyzed, but in the study inclusion criteria were above 16- years of age. It should be change to over 18, and additional statistical analysis should be conducted.
Specifically, taking into account that trampoline injuries are primarily examined in children, we aimed to follow the inclusion criteria implemented in the few conducted studies in adults, that is, above 16-years old, to allow comparability [1]. Similarly, in studies examining trampoline injuries in children the population is defined as under 16-years old [2, 3]. Moreover and in accordance with the imposed policy, children under 16-years of age are admitted to a separate emergency department while patients above 16-years of age are examined in our adult’s emergency department.
We would like to emphasize that we are not in principle opposed to change the inclusion criteria and conduct additional analysis. We have laid out the reasons for which we decided to adopt the specific inclusion criteria and why we believe that this makes our manuscript comparable to the previous literature. If our argument is not found convincing we are happy to adopt it accordingly.
In text citation style should be uninfected. Both are used normal and upper numbers.
Reply: Done as recommended.
Introduction is very short and does not included review of literature: what we already know form previous studies.
Reply: We thank the reviewer for the comment. We have revised the introduction to address the issues raised.
Why the border of group selection was age 30? How authors justify it?
Reply: We thank the reviewer for the opportunity to elaborate on the specific point. Specifically, we used two age groups in order to identify the trends in those specific age groups and to better distinguish between accidents related to school/university
facilities and injuries associated with indoor recreation halls. Our justification is now stated in the manuscript.
104 line in mu opinion p<0.05 instead of p=0.05.
Reply: Done as recommended.
Results. I would recommend instead of 3 similar huge tables presentation only of statistically significant results.
Reply: Done as recommended. Specifically, in the main manuscript we present within the text only the significant results. The complete results in the form of tables are now moved to the Supplementary Material.
In discussion results are repeated but not discussion this other authors was conducted.
Reply: We thank the reviewer for the very important comment. In light of the edifying recommendation additional comments regarding recent literature are now included in the manuscript (Discussion section: Paragraph 3, Paragraph 4, Paragraph 5)
Thank you for the thorough and helpful reviewing and for the opportunity to improve the manuscript. We hope that we have adequately responded to your excellent comments. Having made this revision, we hope that the manuscript would be acceptable for publication in the IJERPH in its current version.
References
- Arora V, Kimmel LA, Yu K, Gabbe BJ, Liew SM, Kamali A. Trampoline related injuries in adults. 2016;47(1):192-196.
- Korhonen L, Niina Salokorpi N, Suo-Palosaari M, Pesälä J, SerloW, Sinikumpu. Severe Trampoline Injuries: Incidence and Risk Factors in Children and Adolescents. Eur J Pediatr Surg. 2018;28(6):529-533.
- Lim F, James V, Pin Lee K , Sashikumar Ganapathy S. A retrospective review of-trampoline-related injuries presenting to a paediatric emergency department in Singapore. Singapore Med J. 2021;62(2):82-86
Round 2
Reviewer 3 Report
The manuscript was significantly improved.
In my opinion to be methodologically correct studied population should be above 18, if authors want to evaluate adult population. Young adult 18-34 could be compared do middle age adults 35-55 (60).
Author Response
Statement Review 3
In my opinion to be methodologically correct studied population should be above 18, if authors want to evaluate adult population. Young adult 18-34 could be compared do middle age adults 35-55 (60).
Answer
We fully understand the reviewer's concern regarding the definition of adult patients in our study. However, the exclusion of patients aged 16-18 would mean the removal of approximately 25% of the cases, which, in our opinion, would significantly reduce the statistical power of the study. We therefore suggest changing the title of the article to "Trampolining accidents in an adult emergency department: analysis of trampolining evolution regarding severity and occurrence of injuries" in order to better summarize and describe the sample and method used. Accordingly, similar modifications have been made to the main manuscript. We would like to emphasize that we are not in principle opposed to exclude the specific age group from the analysis as suggested by the reviewer. If despite our modifications it is still deemed necessary to remove patients between 16 and 18 years of age, we would like to request for an extension of the submission deadline, as all statistical analyses have to be performed from the beginning and, possibly, radical changes to the main manuscript may be required.
In addition, the explanation regarding inclusion of young adult patients older than 16 years of age in adult emergency department in an our publication:
Brockhus LA, Bärtsch M, Exadaktylos AK, Keitel K, Klukowska-Rötzler J, Müller M.(2021) Clinical Presentations of Adolescents Aged 16-18 Years in the Adult Emergency Department. Int J Environ Res Public Health. 2021 Sep 11;18(18):9578. doi: 10.3390/ijerph18189578
Increasing ED use has also been observed in other studies of young adults as
there are often no appropriate primary care services for this physical, developmental, andsocial transition phase. Another reason for this could be that the ED may be the only contact point for adolescents. The UN Convention on the Rights of a Child defines adolescentsas individuals between the ages of 10 and 19 years . This challenge is also present in Switzerland, as there are no dedicated primary care services for adolescents. The adolescent ED population is defined and served differently from country to country
In Switzerland, most EDs have established an artificial cut-off at 16 years and therefore
treat patients aged 16 years or older in the adult ED (Table 1). This approach has
also been adopted at our ED at the University Hospital in Bern.
(Table in the pdf file)
